# Incidence, survival time and associated factors of virological failure among adult HIV/ AIDS patients on first line antiretroviral therapy in St. Paul's Hospital Millennium Medical College—A retrospective cohort study

**Demeke Eshetu Andarge, Haimanot Ewnetu Hailu** ⓞ *◉, **Takele Menna**◉

Department of Public Health, Saint Paul's Hospital Millennium Medical College, Addis Ababa, Ethiopia

◉ These authors contributed equally to this work.
* metiewnetu@gmail.com

## Abstract

### Introduction

Human Immune deficiency Virus or Acquired Immune deficiency Syndrome (HIV/AIDS) is a pandemic affecting millions around the world. The 2020 the Joint United Nations Programme on HIV/AIDS report stated that the estimated number of people living with HIV (PLHIV) is 38 million globally by 2019. Ethiopia is among HIV high burden countries in Africa. By 2021, PLHIV in Ethiopia is estimated to be 754, 256. Globally out of 25.4 million PLHIV on ART, 41% reported virally non-suppressed. According to UNAIDS, the estimated viral non-suppression in Ethiopia is about 27%.

### Methodology

A hospital based retrospective cohort study was conducted among 323 patients who were enrolled to ART from July 2016 to December 2020. The medical records of study participants were selected using simple random sampling technique & data was collected using data extraction checklist. The collected data was entered and cleaned using SPSS V.25. Kaplan–Meier is used to estimate the cumulative hazard of virological failure at different time points. During bivariate analysis variables with p<0.25 were taken for Multivariate Cox regression analysis to assess predictors of virological failure & statistically significant association was declared at p<0.05 with 95% confidence interval.

### Result

The overall incidence rate of virological failure was 1.75 per 1000 months of observations. The mean survival time of virological failure was 14.80 months. Disclosure of sero-status (AHR = 0.038, 95% CI: 0.008–018), poor adherence (AHR = 4.24, 95% CI: 1.04–16), having OIs (Opportunistic infections) (AHR = 4.59, 95% CI: 1.17–18) and use of cotrimoxazole (CPT) prophylaxis (AHR = 0.13, 95% CI: 0.026–0.68) have shown statistically significant association with virological failure.

**Data Availability Statement:** All relevant data are within the paper and its Supporting Information files.

**Funding:** Demeke Eshetu This work was funded by Saint Paul's Hospital Hospital Millennium Medical College. the funders had no role in study design, data collection and analysis, decision to publish, or preparation of the manuscript.

**Competing interests:** The authors have declared that no competing interests exist.

## Conclusion

The incidence of virological failure among patients on first line ART in St. Paul's hospital is low. Disclosure of sero-status, poor adherence, having OIs and use of CPT prophylaxis were associated with virological failure. Therefore, a due attention needs to be given to these factors in order to minimize virological failure in patients on ART.

## Introduction

Human Immune deficiency Virus or Acquired Immune Deficiency Syndrome (HIV/AIDS) is a pandemic affecting millions around the world. The 2020 Joint United Nations Programme on HIV/AIDS (UNAIDS) report stated that the estimated number of people living with HIV was 38 million globally by 2019. Despite the efforts to control this pandemic, the infection rate is high; in 2019, the estimated number of new infections was 1.7million. Of 38 million people living with the Human immune deficiency virus, merely 25.4 million are taking antiretroviral therapy while the remaining are waiting. In 2019, HIV/AIDS claimed life of 690,000 people [1]. About a 74.9million people have been infected with HIV since it became a pandemic and 32 million have died of AIDS-related illness [2].

The vast majority of people living with HIV are located in Low- and Middle- Income Countries (LMIC), with an estimated 68% living in sub-Saharan Africa. Among this group, 20.6 million are living in East and Southern Africa where 800,000 new HIV infections were recorded in 2018 [2].

According to an estimate by the ministry of health, the number of PLHIV in Ethiopia by the years 2020 and 2021 will be 745,719 and 754,256 respectively. Antiretroviral drugs are made available in the country for the first time in 2003 and then free in 2005 [3,4]. UNAIDS has reported that only two-thirds of the 690000 people living with HIV in Ethiopia in 2018 are on treatment [5]. Ethiopian Demographic and Health Survey (EDHS) 2016 indicated that the national prevalence of HIV in Ethiopia was 0.96% [6]. The report from Ethiopia Population-based HIV Impact Assessment (EPHIA) indicates that the 2017–2018 prevalence of HIV among adults aged 15–64 years in urban Ethiopia was 3.0%. In Ethiopia, 81% of all people living with HIV are on treatment and 73% of them were virally suppressed which makes the 27% non-suppressed [7].

The prevalence of HIV/AIDS in Addis Ababa, according to EPHIA, 2017–2018 was 3.1% [7]. The ministry of health estimated people living with HIV in Addis Ababa by the years 2020 and 2021 will be 132,524 and 133,720 respectively [3,4]. The HIV care and treatment service coverage in Addis Ababa indicated 74.6% and viral load testing coverage is about 60% with 87.5% viral suppression among those who received viral load testing [8].

HIV has a lifelong treatment, which is monitored by various means. Viral load (VL) testing is among the mechanisms, which gain wider acceptance these days. Measuring VL can help to distinguish between treatment failure and non-adherence. Studies in 2013 WHO recommended viral load testing [9]. WHO defined virological failure (VF) as, plasma viral load above 1000 copies/ ml based on two consecutive viral load measurements after 3 months, with adherence support [10].

As VL testing is becoming routine across countries, measuring its impact and progress towards achieving the UNAIDS target that 90% of people receiving antiretroviral therapy have suppressed viral loads by 2020 (as part of the 90–90–90 targets) is very important [11].

Viral load suppression can be a performance indicator for ART programs. Regular VL-monitoring allows identification of suboptimal adherence. This recommendation is based on

research demonstrating that viral suppression is associated with decreased HIV disease progression and mortality among people living with the human immune virus (PLHIV), and the prevention of HIV transmission to sexual partners [12,13].

Virological status follow-up will give initial and precise information on the possibility of treatment failure, the necessity to change regimens, lessen mutations that result from drug resistance, and bring desired outcomes. Therefore, VL tests save patients from being needlessly switched to medicines that are more expensive or left to continue on ineffective therapy that can lead to drug resistance and ultimately death [14,15].

The problem has multidimensional consequences on the individual, family, community, economy, and the health system at large. Virological failure is not a sole entity; it goes hand in hand with drug resistance. VF already endangered the handful of drugs that are in use in a fight against the virus. Even though maintaining a low viral load is important for patients to prevent the progression of AIDS and associated co-infections and the rate of HIV infection in Addis Ababa is high, the evidence on virological failure, survival time and the associated factors are limited. Therefore, this study is aimed to fill this gap in producing evidence that can be useful in making an informed decision. In addition, it will contribute to the realization of the 90-90-90 treatment target and achieve sustainable development goal 3.

## Method and materials

A retrospective cohort study was conducted on HIV infected participants on a WHO recommended antiretroviral therapy (ART) and enrolled in Saint Paul's Hospital Millennium medical college between July 2016 and December 2020.

St. Paul's Hospital Millennium Medical College is located in Addis Ababa, the capital. The ART service was started in 2003 and currently more than 5080 clients are getting the service free of charge.

Sample size was determined using Epi Info™ Version: 7.2.1.0 StatCalc by considering 95% confidence level, 80% power, unexposed to exposed group ratio of one and taking the key predictor of VF (BMI <16 which gave the largest sample size among the variables) from a previous study in Woldia and Dessie hospitals and Waghimra zone. Therefore, the calculated minimum sample size is 308 and by considering a 10% non-response rate the final sample is 340.

The outcome variable was virological failure is plasma viral load above 1000 copies/ ml based on two consecutive viral load measurements after 3 months, with adherence support. The potential associated factors included age, gender, education, marital status, disclosure status, occupational status, BMI, base line drug regimen, cotrimoxazole prophylaxis, base line functional status, WHO stage, adherence to treatment, TB/HIV co-infection, opportunistic infections other than TB and CD4 cell count. A simple random sampling technique was used to select participants who are greater than 15 years of age and on ART at least for 10 months before data collection. Data was extracted from patient cards using a structured checklist prepared in English adapted from Ethiopian Federal Ministry of Health ART clinic intake and follow up form.

Inclusion and exclusion criteria: The inclusion criteria include: clients aged 15years or older, who were on treatment for at least ten months and on first line ART. Transfer in and those without viral load test were excluded from the study.

Operational definition: Censored are those patients complete the follow up, transferred out or lost without developing virological failure. Adherence is defined as follows: Clients on ART with >95% adherence are considered to have good adherence, those with 85–94% adherence are fair and those with <85% are considered to have poor adherence [16]. History of

Opportunistic infection or Tuberculosis indicates whether the patient has history of any Opportunistic infection including Tuberculosis. On the other hand, recent infection is whether the person currently has TB or any Opportunistic infection.

Data were entered, cleared and analyzed using SPSS version 25. Descriptive statistics was used to describe demographic, clinical and medication-related characteristics of patients. The Kaplan-Meier method was used to estimate the cumulative incidence of virological failure at different time points. Incidence of virological failure was calculated using Person months (PM) observation. Cox proportional hazards model was used to identify factors significantly associated with virological failure and to control confounding factors. To control the confounders, a multivariable model was developed for a priori confounders including age and sex which are selected based on existing literature. A p- value of less than 0.05 with 95% was used to declare statistical significance.

This study was approved by the SPHMMC Institutional review Board. An official letter of permission was obtained from the Hospital to access the data from the record of patients that is fully anonymized before we accessed them. An informed verbal consent was obtained from the study participants. The obtained information was kept confidential and only be used for research purpose.

## Result

### Socio-demographic characteristics

From July 2016 to December 2020, 640 adult HIV patients on first-line ART were enrolled in St. Paul's hospital millennium medical college ART clinic and 340 medical record cards were selected using simple random sampling of which 17 medical record cards were excluded due to missed charts and incomplete data. As a result, a total of 323 patient cards were included in the analysis

The mean age of the patients was 36.86 (SD±9.8) with minimum age of 16 and maximum 70 years. More than half, 188 (58.2%) of the patients were female. One hundred seven (33.1%) attended primary level of education and 123(38.1%) were with secondary education. Half of the participant in this study 163(50.5%) were married and 142(43.7%) were employed. More than 3/4[th] of the participants disclosed their HIV status at least for one person (Table 1).

### Baseline clinical and anti-retroviral medication-related characteristics

Majority of the participants 202(62.5%) had normal body mass index. Two hundred sixty nine, 269(83.3%) of the total participants were on Efavernez (EFV) based first line ART drug regimen. More than half of the patients, 174(53.9%) took Cotrimoxazole preventive therapy (CPT). Nearly all 320(99.1%) patients in the study could perform their routine activities. More than half of the patients 193(59.8%) had baseline WHO clinical stage I/II and 247(76.5%) had good ART adherence status. On the other hand, 46(14.2) had TB/HIV co-infection and 82 (25.4%) experienced OI other than TB. More than one third (36.5%) had a base line CD4 count of less than 200 copies ml and the mean month of developing virological failure was 30 (SD±12) with minimum 10.4 and maximum 53.26 months respectively (Table 2).

### The incidence of virological failure

All the participants of the study were followed for different periods with a total of 9,698.36 person-months (PM) of observations. The first patient developed the event (VF) after 10.4 months of follow up and the last after 53.3 months. Seventeen, 17(5.3%) of patients developed VF during the follow-up period. The overall incidence rate of VF in this follows up was 1.75 events

**Table 1. Base line Socio-demographic characteristics of first line ART clients in St. Paul's hospital millennium medical college in Addis Ababa, Ethiopia from July 2016 to December 2020 (N = 323).**

| Variable | Frequency (%) |
|---|---|
| Age group | |
| under 20 | 9(2.8) |
| 21–30 | 92(28.5) |
| 31–40 | 121(37.5) |
| 41–50 | 74(22.9) |
| 51 and above | 27(8.3) |
| Gender | |
| female | 188(58.2) |
| male | 135(41.8) |
| Education | |
| No formal education | 37(11.5) |
| Primary | 107(33.1) |
| Secondary | 123(38.1) |
| College and above | 56(17.3) |
| Marital status | |
| Married | 163(50.5) |
| Never married | 142(44) |
| Divorced/widowed | 18(5.6) |
| Disclosure status | |
| Disclosed | 252(78) |
| Not disclosed | 57(17.6) |
| unknown | 14(4.3) |
| Occupational status | |
| employed | 141(43.7) |
| self employed | 42(13.0) |
| unemployed | 132(40.9) |
| Others | 8(2.5) |

per 1000 PM of observations. The cumulative hazard of VF at 12, 24, 36 and 48 months were 1.2%, 3.7%, 5.5% and 7.6% respectively.

A graph of the Kaplan Meier (KM) failure function was used to describe the cumulative IR of virological failure over the follow-up period (Fig 1).

## Survival time of virological failure

The cumulative probability of surviving or being free from the event of interest, VF at the end of 12, 24, 36 and 48 months was 98.76%, 96.30%, 94.49% and 92.35% respectively (Fig 2). The mean survival time of virological failure was 14.8 months.

## Factors associated with virological failure

The risk of developing VF among participants disclosed their HIV status decreased by 96.2% compared to their counterparts (AHR) (AHR = 0.038, 95% CI: 0.008–0.18). Patients with poor treatment adherence were four times more likely to develop VF compared to those with good adherence (AHR = 4.24, 95% CI: 1.04–16). Similarly, patients having history of OIs were at four-fold risk of VF compared to those without OIs (AHR = 4.59, 95% CI: 1.17–18). In addition, patients taking cotrimoxazole prophylaxis are 87% less likely to develop to VF compared to those who are not on the prophylaxis (AHR = 0.13, 95% CI: 0.026–0.68) (Table 3).

**Table 2. Baseline clinical and antiretroviral medication-related information among adult HIV patients on first-line ART in St. Paul's hospital millennium medical college in Addis Ababa, Ethiopia from July 2016 to December 2020 (N = 323).**

| Variable | Frequency (%) |
|---|---|
| BMI category (Kg/m$^2$) | |
| under18.5 | 44(13.6) |
| 18.5–24.9 | 202(62.5) |
| 25–29.9 | 65(20.1) |
| 30 and above | 12(3.7) |
| Base line drug regimen | |
| Nevirapine based | 11(3.4) |
| Efavirenz based | 269(83.3) |
| DTG based | 43(13.3) |
| Cotrimoxazole prophylaxis* | |
| yes | 174(53.9) |
| no | 149(46.1) |
| Base line functional status | |
| Working | 320(99.1) |
| Ambulatory | 3(0.9) |
| WHO stage | |
| I/II | 193(59.8) |
| III | 79(24.5) |
| IV | 51(15.8) |
| Adherence to treatment | |
| Fair/poor | 76(23.5) |
| Good | 247(76.5) |
| TB/HIV co-infection | |
| yes | 46(14.2) |
| no | 277(85.8) |
| OIs other than TB | |
| yes | 82(25.4) |
| no | 241(74.6) |
| CD4 (cells/mm$^3$) | |
| 200 and below | 118(36.5) |
| 201–350 | 75(23.2) |
| 351–500 | 52(16.1) |
| 501 and above | 60(19.7) |
| Missing** | 18(5.6) |

*taken cotrimoxazole prophylaxes at any time in the follow up time.

## Discussion

This study assessed the incidence rate of VF and associated factors among adult HIV/AIDS patients on first- line ART attending St. Paul's hospital millennium medical college in Addis Ababa, Ethiopia.

The overall incidence rate of VF in this study was 1.75 per 1000 PM of observation (95 CI: 0.024–0.068). This result was lower than a finding of similar study conducted in Amhara referral hospital, which was 4.9 per 1000 PM [16]. The reason for discrepancy may be the high proportion of underweight patients in the Amhara hospital study as compared to the current

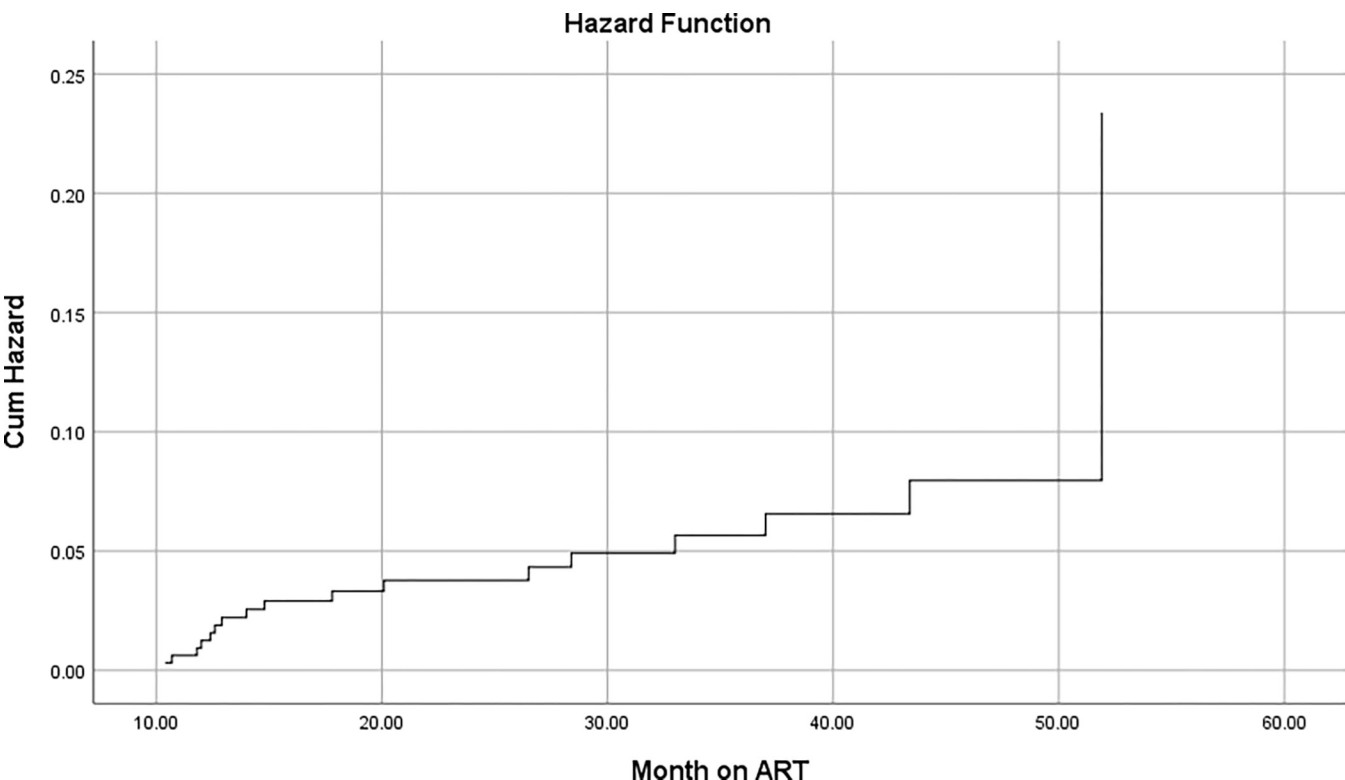

**Fig 1. Cumulative incidence of VF among first line adult ART patients in St. Paul's hospital millennium medical college from July 2016 to December 2020.**

study that is 32.24% vs 13.6%. Studies illustrated that underweight clients on ART are at more risk of VF [17]. In addition, the study in Amhara region hospital reported that proportion of patients with no formal education and primary education was 51% higher compared to the current study, which was 44%, and studies suggested that patients with lower educational level are at higher risk of developing VF [18]. The socio-cultural difference between the study settings may be another contributing factor for the difference in the findings. Majority of the participants of this study were dwellers of the capital, Addis Ababa, with better access to service and information, which might have a contribution for lower Virological Failure.

A similar study in Adama reported the incidence rate of 2.1 per 1000 PM, which was more or less comparable with the result of current study [19]. This could be the physical proximity between the two cities and socio-cultural similarities.

A study in South Africa indicated 3.8 events per 1000 PM observation which was higher than this study [20]. The cohort in South Africa took patients on treatment for a relatively longer duration compared to this study. As indicated in some studies the increase in treatment duration increases the risk of developing VF. The possible explanation for this includes tolerance developed by the patients against advices by professionals, pill fatigue and/ or mutation of the virus over time [21,22]. The other possible reason that caused the discrepancy could be the large sample size used in the South African study.

The incidence rate reported from Ugandan study was 4.8 per 1000 PM, which was higher than the finding in our study [23]. The residence area of the study participants could be an explanation for the observed difference. More than three fourth (76.8%) of the participants were from rural areas of Uganda as opposed to the current study of which majority of the

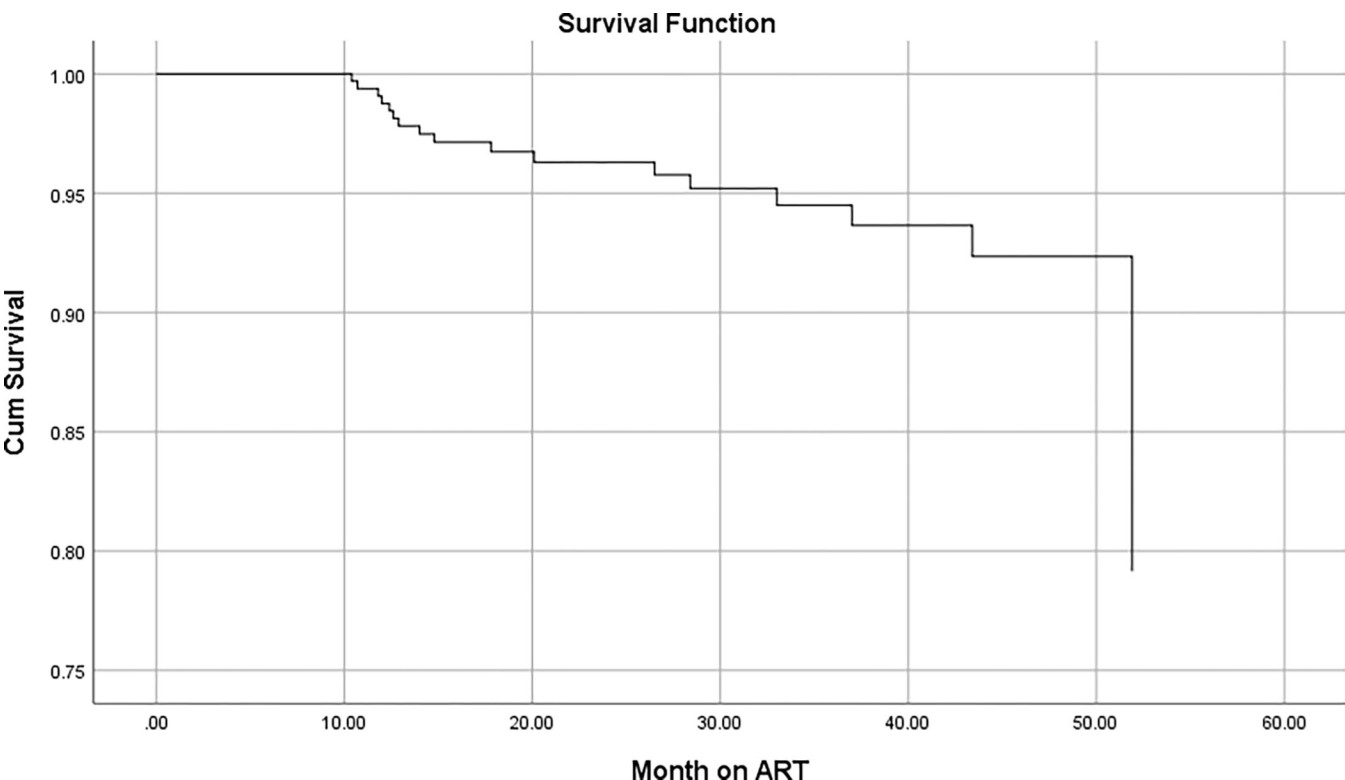

**Fig 2. Kaplan-Meier's survival graph of patients on ART in St. Paul's hospital millennium medical college from July 2016 to December 2020.**

patients were from the capital. This could be explained by the fact that urban residents have better awareness of the treatment and the risk for VF is lesser. The other possible reason could be that the age of the study participants; where 44.1% of the Ugandan cohort were the below the age of 30 years while the proportion of the same age group in current study was 31.3%. As different articles suggest the young age group is highly vulnerable to VF [19,24,25].

Another similar study from India reported a VF incidence rate of 8.9 event per 1000 PM observation [26]. The cause for variation of the two results was the study in India excluded those patients with higher CD4>350 and WHO stage II and I. Patients with advanced WHO staging and lower CD4 count are at risk of developing VF [14,27,28].

The finding of the current study exhibited that the mean survival time of virological failure was 14.8 months. This could be due to the fact that, when clients start a treatment there might be issues with acceptance of the treatment and intolerance to adverse effects may develop which this in turn affects adherence and subsequent development of virological failure.

The study revealed that participants who disclosed their HIV status were 96.2% less likely to develop virological failure. The possible explanation for this could be that, ART is a lifelong treatment and care that demand the support of others. Those who disclosed their status could get the necessary moral, psychological and material support when they are in need in contrast to their counterparts. Patients who declared their HIV status to their loved ones and friends can freely take their medication in front of family members, even reminded by family members and friends to take drugs on time. This study finding is in line with studies in Zimbabwe [29] and Uganda [23].

The other factor found to be significantly associated with VF is poor adherence. Patients with poor adherence are more than 4 times at risk of virological failure than those with good

**Table 3. Factors associated with virological failure among first ART clients in St. Paul's Hospital Millennium Medical College, Addis Ababa, Ethiopia from July 2016 to December 2020.**

| Variable | Event | Censored | P-value | CHR 95% CI | P-value | AHR 95% CI |
|---|---|---|---|---|---|---|
| Gender | | | | | | |
| Male | 12 | 123 | **0.023** | 3.35(1.18–9.51) | 0.73 | 1.26(0.35–4.59) |
| Female | 5 | 283 | | 1 | | 1.000 |
| Education status | | | | | | |
| No formal education | 2 | 35 | 0.31 | 0.44(0.09–2.2) | 0.79 | 1.38(0.12–16.55) |
| Primary | 3 | 104 | **0.04** | 0.22(0.05–0.91) | 0.16 | 0.19(0.019–1.89) |
| Secondary | 6 | 117 | 0.09 | 0.37(0.12–1.2) | 0.52 | 0.56(0.09–3.22) |
| college & above | 6 | 50 | | 1 | | 1.000 |
| Base line functional status | | | | | | |
| Working | 16 | 304 | **0.069** | 0.15(0.020–1.16) | 0.96 | 0.92(0.039–21) |
| Ambulatory | 1 | 2 | | 1 | | 1.000 |
| Marital status | | | | | | |
| Married | 7 | 156 | **0.21** | 0.36(0.07–1.74) | 0.75 | 1.52(0.12–19.6) |
| Never married | 8 | 134 | 0.36 | 0.48(0.1–2.3) | 0.68 | 1.75(0.13–23.85) |
| Divorced/widowed | 2 | 16 | | 1 | | 1.000 |
| Disclosure status | | | | | | |
| Disclosed | 6 | 246 | **0.000** | 0.11(0.04–0.29) | **0.000** | 0.038(0.008–0.18) |
| Not disclosed | 11 | 46 | | 1 | | 1 |
| Adherence to treatment | | | | | | |
| Fair/poor | 11 | 65 | **0.00** | 6.3(2.31–17.09) | **0.040** | 4.24(1.06–16.9) |
| Good | 6 | 241 | | 1 | | 1 |
| WHO stage of participant | | | | | | |
| WHO stage I/II | 2 | 191 | **0.000** | 0.041(0.009–0.185) | 0.27 | 0.29(0.03–2.62) |
| WHO stage III | 3 | 76 | 0.004 | 0.15(0.043–0.55) | 0.18 | 0.29(0.05–1.77) |
| WHO stage IV | 12 | 39 | | 1 | | 1.000 |
| TB/HIV co-infection | | | | | | |
| Yes | 9 | 37 | **0.000** | 7.23(2.77–18.8) | 0.11 | 3.07(0.76–12.36 |
| No | 8 | 269 | | 1 | | 1.000 |
| OIs other than TB | 8 | 269 | | 1 | | 1.000 |
| Yes | 12 | 70 | **0.000** | 6.97(2.45–19.7) | **0.029** | 4.59(1.17–18.07) |
| No | 5 | 236 | | 1 | | 1.000 |
| Cotrimoxazole prophylaxes (CPT) | | | | | | |
| Yes | 11 | 163 | 0.18 | 0.14(0.53–3.9) | **0.015** | 0.13(0.026–0.68) |
| No | 6 | 143 | | 1 | | 1 |
| CD4 count | | | | | | |
| CD4 = <200 | 14 | 104 | **0.051** | 4.45(0.99–19.9) | 0.216 | 3.56(0.476–26.66) |
| CD4 201–350 | 1 | 74 | 0.490 | 0.44(0.040–4.8) | 0.88 | 0.82(0.062–10.82) |
| CD4 351–500 | 0 | 52 | 0.970 | 0 | 0.919 | 0.000 |
| CD4 > = 501 | 2 | 58 | | 1 | | 1.000 |

adherences. This finding is in agreement with studies in India [26], Addis Ababa [8] Mekele [14], Gondar [21], Dessie and Woldia [27], Adama [19] and Amhara regional hospitals [16]. In fight against infections like HIV, adhering to the treatment is a key, otherwise it creates a situation that leads to development of resistant varieties that cannot be controlled by the medications at hand. Patients with poor adherence are exposed to CD4 cell reduction, which in turn affects their immune status, and rise viral load [20].

Similarly, the hazard of VF of patients with OIs is 4 times as compared to patients without OIs. Some studies indicate that the presence of other infections with HIV could affect the ART care service [30,31]. Unfortunately, OIs are common among HIV patients. Some OIs are recurrent in nature and made the patient take multiple drugs, this in turn made patient to focus on current illness and give less attention to the chronic condition and pill fatigue may occur. When patient overwhelmed by pill burden they fail to take their ART properly leading to VF [32].

Patients on CPT prophylaxis were 86.6% less likely to develop VF compared to those who did not take the prophylaxis. This happens because CPT increases the CD4 cells of patients and improves their immune status. When the immune system of the patients improve the viral multiplication decreases making them at lower risk of VF [16].

Limitations of this study were, use of secondary data, socio-economic factors like income, behavioral factors like alcohol consumption, smoking and drug use and psychological factors like depression were not included. In addition, though multivariable analyses were adjusted for a known confounder, residual confounding factors cannot be ruled out and the results should be interpreted with caution.

## Conclusion

The incidence rate of virological failure was low among HIV patients on the first line ART at St. Paul's hospital millennium medical college. Higher level of virological failure is observed at early stage of the treatment. The mean survival time of virological failure was 14.8 months. Poor adherence to treatment, failure to disclose HIV status, having OIs and not using cotrimoxazole prophylaxis were the factors associated with increased risk of virological failure. Due attention needs to be given for patients with this conditions during the follow up time.

## Supporting information

**S1 Data.**
(SAV)

## Acknowledgments

We thank the Saint Paul's Hospital Hospital Millennium Medical College ART clinic for their cooperation.

## Author Contributions

**Conceptualization:** Demeke Eshetu Andarge.

**Data curation:** Demeke Eshetu Andarge.

**Formal analysis:** Demeke Eshetu Andarge, Haimanot Ewnetu Hailu, Takele Menna.

**Funding acquisition:** Demeke Eshetu Andarge.

**Investigation:** Demeke Eshetu Andarge, Takele Menna.

**Methodology:** Demeke Eshetu Andarge, Haimanot Ewnetu Hailu, Takele Menna.

**Project administration:** Demeke Eshetu Andarge, Haimanot Ewnetu Hailu.

**Resources:** Demeke Eshetu Andarge, Haimanot Ewnetu Hailu, Takele Menna.

**Software:** Demeke Eshetu Andarge, Haimanot Ewnetu Hailu, Takele Menna.

**Supervision:** Haimanot Ewnetu Hailu, Takele Menna.

**Validation:** Haimanot Ewnetu Hailu, Takele Menna.

**Visualization:** Demeke Eshetu Andarge, Haimanot Ewnetu Hailu.

**Writing – original draft:** Demeke Eshetu Andarge.

**Writing – review & editing:** Haimanot Ewnetu Hailu, Takele Menna.

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
