## [Decision Letter · Decision Letter 0]

24 Nov 2021

PONE-D-21-20837Incidence, survival time and associated factors of virological failure among adult HIV/AIDS patients on first line antiretroviral therapy in St. Paul’s Hospital Millennium Medical College - a retrospective cohort studyPLOS ONE

Dear Dr. Hailu,

Thank you for submitting your manuscript to PLOS ONE. After careful consideration, we feel that it has merit but does not fully meet PLOS ONE’s publication criteria as it currently stands. Therefore, we invite you to submit a revised version of the manuscript that addresses the points raised during the review process.

We look forward to receiving your revised manuscript.

Kind regards,

Giordano Madeddu

Academic Editor

PLOS ONE

Journal Requirements:

3. We note that you have stated that you will provide repository information for your data at acceptance. Should your manuscript be accepted for publication, we will hold it until you provide the relevant accession numbers or DOIs necessary to access your data. If you wish to make changes to your Data Availability statement, please describe these changes in your cover letter and we will update your Data Availability statement to reflect the information you provide

4. We noticed you have some minor occurrence of overlapping text with the following previous publication(s), which needs to be addressed:

- https://journals.plos.org/plosone/article?id=10.1371%2Fjournal.pone.0196259

The text that needs to be addressed involves the Introduction.

In your revision ensure you cite all your sources (including your own works), and quote or rephrase any duplicated text outside the methods section. Further consideration is dependent on these concerns being addressed.

Reviewers' comments:

Reviewer's Responses to Questions

**Comments to the Author**

1. Is the manuscript technically sound, and do the data support the conclusions?

Reviewer #1: Partly

Reviewer #2: Yes

2. Has the statistical analysis been performed appropriately and rigorously? 

Reviewer #1: No

Reviewer #2: Yes

3. Have the authors made all data underlying the findings in their manuscript fully available?

Reviewer #1: No

Reviewer #2: Yes

4. Is the manuscript presented in an intelligible fashion and written in standard English?

Reviewer #1: No

Reviewer #2: Yes

5. Review Comments to the Author

Reviewer #1: This study is a retrospective cohort analytic study of virological failure among HIV adult patients on first-line drug treatment and its factor relation in St. Paul's Hospital Millennium Medical College, Ethiopia. These are the comments on this manuscript.

Major comment

1. The result of this study was not a novel finding. Many studies from Ethiopia and in extensive Africa cohort studies explored the several impact factors associated with virological failure (VF), which should add in reference.

2. The methodology was not clear and could not follow by the STROBE checklist. For example: how to study size determined, how the loss to follow-up handled, how and when should be the censor, and how many numbers of completely follow up were analyzed.

3. In table 3, How to report the number of outcome events or summary measures over time.

4. The study duration was July 2016-Dec 2019, but the patient who last developed the event was 53 months, and VF reported at the end of 48 months that were not consistent with the duration of the study.

5. Statistical method was not precise which confounders were chosen to adjust for and why or exploratory multivariate survival was analyzed.

6. Reference 20 could not be comparable in contextualized higher incidence because that previous study was done in naive patients with poor CD4 and high viral load.

7. This manuscript is needed to clarify many points such as good adherence definition, OI/TB former history, or recent infection.

8. In conclusion, Cotrimoxazole prophylaxes(CPT) are given or not depend on the CD4 level. CTP plus antiretroviral (ARV) drug could decrease viral load in cases of an initial high viral load. This study should examine subgroups and interactions among CPT, CD4 level, and ARV.

9. The manuscript's title declared patient used the first-line drug, but the inclusion criteria had not inferred the antiretroviral type. The manuscript should be more clear.

10. Some typo errors and English language in the discussion part might be made readers confused. This study should examine subgroups and interactions among CPT, CD4 level, and ARV.

9. The manuscript's title declared that the patient used the first-line drug but did not infer the antiretroviral type in the inclusion criteria. The manuscript should be more precise.

10. Some typo errors and English language in the discussion part might be made readers confused.

Reviewer #2: PEER REVIEW

Title – Incidence, survival time and associated factors of virological failure among adult HIV/AIDS patients on first line antiretroviral therapy in St. Paul’s Hospital Millennium Medical College - a retrospective cohort study

Authors - Demeke Eshetu Andarge, Haimanot Ewnetu Hailu, Takele Menna

1.0 Summary of the Research and Overall Impression

The research paper by Demeke et al is a retrospective cohort study that was conducted at the St. Paul’s Hospital Millennium Medical College between 2016 to 2019. The study examines 323 HIV/AIDS adult patients on first line antiretroviral (ART) patients enrolled on ART for virological failure. Data was collected using random sampling from medical records. Virological failure was defined as plasma viral load above 1000 copies/ ml based on two consecutive viral load measurements after 3 months, with adherence support. Data on treatment and preventive therapy, CD4 cell count, adherence, coinfection (Opportunistic Infections (OI) and Tuberculosis (TB), body mass index, age, sex, marital status, education level and employment. In addition, patients were categorized based on WHO clinical stage I/II. Analysis was done using Multivariate Cox regression analysis to assess predictors of virological failure and statistically significant at p < 0.05 with 95% confidence intervals were reported. Kaplan–Meier was used to estimate virological failure. The overall incidence rate of VF was 1.75 events per 1000 PM of observations. The mean survival time of virological failure was 14.8 months. Factors associated with virological failure were poor adherence, non-disclosure of HIV status, no prophylaxis use, and OI. More females (58.2%) had viral failure then men, there were slightly more patients with secondary education 38.1% as opposed to primary level of education. Employment and marital status were equally distributed among patients. More than 75% of patients disclosed their status and the mean age for failure was 36.86.

Overall, this paper contributes important information that supports the existing literature on this topic. All relevant data is provided, and the statistical analysis is appropriate.

The research paper provides evidence on Incidence and survival time for virological failure among adult HIV/AIDS on first line antiretroviral therapy. This research is timely, as it provides information that will assist with the determination of the associated factors of virological failure among adult HIV/AIDS patients on antiretroviral therapy in St. Paul’s Hospital Millennium Medical College. The information has implication for the last 90 of the 2020 UNAIDs 9-90-90 target (90% of all people living with HIV will know their HIV status, 90% of all people with diagnosed HIV infection will receive sustained treatment, and 90% of all people receiving treatment will have viral load suppression). The evaluation of impact of treatment strategy on viral load and disease stage by gender, age, marital status, education level and employment provide important information that can be used to identify areas for intervention during follow up to improve outcome.

It was noted that in the document the author states that the use of CPT prophylaxis was associated with virological failure. The statement in the Results section, Paragraph 5, Factors Associated with Virological Failure, Line 6 “patients taking cotrimoxazole prophylaxis are 87% less likely to develop to VF compared to those who are on the prophylaxis” this statement was not clear. The authors should clarify to avoid confusion and ensure that this statement on results support the conclusion.

2.0 Discussion of specific areas for improvement

Evidence and Examples

Major Issues - There are no major issues to be addressed

Minor Issues -The authors should clarify the following section to avoid confusion

Results- Paragraph 5 - Factors associated with virological failure- Line 6

1. “patients taking cotrimoxazole prophylaxis are 87% less likely to develop to VF compared to those who are on the prophylaxis”

6. PLOS authors have the option to publish the peer review history of their article (what does this mean?). If published, this will include your full peer review and any attached files.

Reviewer #1: No

Reviewer #2: No

---

## [Author Response · Author response to Decision Letter 0]

4 Jan 2022

We would like to thank the reviewers for their unreserved support. All the comments has been addressed accordingly.

---

## [Decision Letter · Decision Letter 1]

13 Jul 2022

PONE-D-21-20837R1Incidence, survival time and associated factors of virological failure among adult HIV/AIDS patients on first line antiretroviral therapy in St. Paul’s Hospital Millennium Medical College - a retrospective cohort studyPLOS ONE

Dear Dr. Hailu,

Thank you for submitting your manuscript to PLOS ONE. After careful consideration, we feel that it has merit but does not fully meet PLOS ONE’s publication criteria as it currently stands. Therefore, we invite you to submit a revised version of the manuscript that addresses the points raised during the review process.

Thank you for revising your manuscript to respond to the reviewer's comments. One of the previous reviewers has reviewed the manuscript and considers it suitable for publication, pending a small number of changes that you will see below. The other reviewer was not able to asses your submission, so I have reviewed the response to their comments myself. I have also assessed the manuscript. The following modifications are required for publication:

1) The study dates in the main body of the manuscript were previously updated, but have not been updated in the Abstract - please revise accordingly.

2) The previous reviewer requested further detail on how confounding variables were identified. This concern has not been fully addressed; please update your Methods section to outline how potential confounding variables were identified.

3) Related to the above, please ensure that all variables reported in the Results section are specified in the Methods section and provide details of how each variable was measured.

We look forward to receiving your revised manuscript.

Kind regards,

George Vousden

Deputy Editor in Chief

PLOS ONE

Journal Requirements:

Reviewers' comments:

Reviewer's Responses to Questions

**Comments to the Author**

1. If the authors have adequately addressed your comments raised in a previous round of review and you feel that this manuscript is now acceptable for publication, you may indicate that here to bypass the “Comments to the Author” section, enter your conflict of interest statement in the “Confidential to Editor” section, and submit your "Accept" recommendation.

Reviewer #2: All comments have been addressed

2. Is the manuscript technically sound, and do the data support the conclusions?

Reviewer #2: Yes

3. Has the statistical analysis been performed appropriately and rigorously? 

Reviewer #2: Yes

4. Have the authors made all data underlying the findings in their manuscript fully available?

Reviewer #2: Yes

5. Is the manuscript presented in an intelligible fashion and written in standard English?

Reviewer #2: Yes

6. Review Comments to the Author

Reviewer #2: PEER REVIEW

Title – Incidence, survival time and associated factors of virological failure among adult HIV/AIDS patients on first line antiretroviral therapy in St. Paul’s Hospital Millennium Medical College - a retrospective cohort study

Authors - Demeke Eshetu Andarge, Haimanot Ewnetu Hailu, Takele Menna

1.0 Summary of the Research and Overall Impression

The research paper by Demeke et al is a retrospective cohort study that was conducted at the St. Paul’s Hospital Millennium Medical College between 2016 to 2019. The study examines 323 HIV/AIDS adult patients on first line antiretroviral (ART) patients enrolled on ART for virological failure. Data was collected using random sampling from medical records. Virological failure was defined as plasma viral load above 1000 copies/ ml based on two consecutive viral load measurements after 3 months, with adherence support. Data on treatment and preventive therapy, CD4 cell count, adherence, coinfection (Opportunistic Infections (OI) and Tuberculosis (TB), body mass index, age, sex, marital status, education level and employment. In addition, patients were categorized based on WHO clinical stage I/II. Analysis was done using Multivariate Cox regression analysis to assess predictors of virological failure and statistically significant at p < 0.05 with 95% confidence intervals were reported. Kaplan–Meier was used to estimate virological failure. The overall incidence rate of VF was 1.75 events per 1000 PM of observations. The mean survival time of virological failure was 14.8 months. Factors associated with virological failure were poor adherence, non-disclosure of HIV status, no prophylaxis use, and OI. More females (58.2%) had viral failure then men, there were slightly more patients with secondary education 38.1% as opposed to primary level of education. Employment and marital status were equally distributed among patients. More than 75% of patients disclosed their status and the mean age for failure was 36.86.

Overall, this paper contributes important information that supports the existing literature on this topic. All relevant data is provided, and the statistical analysis is appropriate.

The research paper provides evidence on Incidence and survival time for virological failure among adult HIV/AIDS on first line antiretroviral therapy. This research is timely, as it provides information that will assist with the determination of the associated factors of virological failure among adult HIV/AIDS patients on antiretroviral therapy in St. Paul’s Hospital Millennium Medical College. The information has implication for the last 90 of the 2020 UNAIDs 9-90-90 target (90% of all people living with HIV will know their HIV status, 90% of all people with diagnosed HIV infection will receive sustained treatment, and 90% of all people receiving treatment will have viral load suppression). The evaluation of impact of treatment strategy on viral load and disease stage by gender, age, marital status, education level and employment provide important information that can be used to identify areas for intervention during follow up to improve outcome.

It was noted that in the document the author states that the use of CPT prophylaxis was associated with virological failure. The statement in the Results section, Paragraph 5, Factors Associated with Virological Failure, Line 6 “patients taking cotrimoxazole prophylaxis are 87% less likely to develop to VF compared to those who are on the prophylaxis” the statement is contradictory. The authors should clarify to avoid confusion and ensure that the statement of results support the conclusions.

2.0 Discussion of specific areas for improvement

Evidence and Examples

Major Issues

There are no major issues to be addressed

Minor Issues

The authors should clarify the following sections to avoid confusion

Results- Paragraph 5 - Factors associated with virological failure- Line 6

1. “patients taking cotrimoxazole prophylaxis are 87% less likely to develop to VF compared to those who are on the prophylaxis”

Response to Review - April 4, 2022

Response: Thank you for your comment. It is well addressed. It is typo error. It was corrected as follows: Patients taking cotrimoxazole prophylaxis are 87% less likely to develop to VF compared to those who are NOT on the prophylaxis (AHR=0.13, 95% CI: 0.026- 0.68).

Correction noted and accepted in the research paper.

7. PLOS authors have the option to publish the peer review history of their article (what does this mean?). If published, this will include your full peer review and any attached files.

Reviewer #2: No

---

## [Author Response · Author response to Decision Letter 1]

19 Aug 2022

Thank you for your feedback. All the comments are well addressed and included in the manuscript and response to reviewers section.

---

## [Editor Report · Decision Letter 2]

1 Sep 2022

PONE-D-21-20837R2Incidence, survival time and associated factors of virological failure among adult HIV/AIDS patients on first line antiretroviral therapy in St. Paul’s Hospital Millennium Medical College - a retrospective cohort studyPLOS ONE

Dear Dr. Hailu,

Thank you for submitting your manuscript to PLOS ONE. After careful consideration, we feel that it has merit but does not fully meet PLOS ONE’s publication criteria as it currently stands. It remains unclear how confounding factors were identified; the Methods section only indicates that "Cox proportional hazards model was used to identify factors significantly associated with virological failure and to control confounding factors.". The specific methods to identify potentially confounding factors are not clear from this sentence. Please provide further detail.

We look forward to receiving your revised manuscript.

Kind regards,

George Vousden

Deputy Editor in Chief

PLOS ONE
---

## [Author Response · Author response to Decision Letter 2]

3 Sep 2022

Thank you for your comment. The details are addressed in the methods and limitation section too.

---

## [Editor Report · Decision Letter 3]

13 Sep 2022

Incidence, survival time and associated factors of virological failure among adult HIV/AIDS patients on first line antiretroviral therapy in St. Paul’s Hospital Millennium Medical College - a retrospective cohort study

PONE-D-21-20837R3

Dear Dr. Hailu,

We’re pleased to inform you that your manuscript has been judged scientifically suitable for publication and will be formally accepted for publication once it meets all outstanding technical requirements.

Kind regards,

George Vousden

Staff Editor

PLOS ONE
---

## [Editor Report · Acceptance letter]

2 Oct 2022

PONE-D-21-20837R3 

Incidence, survival time and associated factors of virological failure among adult HIV/AIDS patients on first line antiretroviral therapy in St. Paul’s Hospital Millennium Medical College - a retrospective cohort study 

Dear Dr. Hailu:

I'm pleased to inform you that your manuscript has been deemed suitable for publication in PLOS ONE. Congratulations! Your manuscript is now with our production department. 

Kind regards, 

on behalf of

Dr. George Vousden 

Staff Editor

PLOS ONE